# Lattice-mismatch-free growth of organic heterostructure nanowires from cocrystals to alloys

Qiang Lv[1,2], Xue-Dong Wang [1✉], Yue Yu[1], Ming-Peng Zhuo[1], Min Zheng [2,3✉] & Liang-Sheng Liao [1,4✉]

Organic heterostructure nanowires, such as multiblock, core/shell, branch-like and related compounds, have attracted chemists' extensive attention because of their novel physico-chemical properties. However, owing to the difficulty in solving the lattice mismatch of distinct molecules, the construction of organic heterostructures at large scale remains challenging, which restricts its wide use in future applications. In this work, we define a concept of lattice-mismatch-free for hierarchical self-assembly of organic semiconductor molecules, allowing for the large-scale synthesis of organic heterostructure nanowires composed of the organic alloys and cocrystals. Thus, various types of organic triblock nanowires are prepared in large scale, and the length ratio of different segments of the triblock nanowires can be precisely regulated by changing the stoichiometric ratio of different components. These results pave the way towards fine synthesis of heterostructures in a large scale and facilitate their applications in organic optoelectronics at micro/nanoscale.

[1] Institute of Functional Nano & Soft Materials (FUNSOM), Jiangsu Key Laboratory for Carbon-Based Functional Materials & Devices, Soochow University, 199 Ren'ai Road, Suzhou, Jiangsu 215123, PR China. [2] National Engineering Laboratory for Modern Silk, College of Textile and Clothing Engineering, Research Center of Cooperative Innovation for Functional Organic/Polymer Material Micro/Nanofabrication, Soochow University, Suzhou, Jiangsu 215123, PR China. [3] Jiangsu Naton Science&technology Co., Ltd, Suzhou Industrial Park, Suzhou, Jiangsu 215123, PR China. [4] Macao Institute of Materials Science and Engineering, Macau University of Science and Technology, Taipa, 999078 Macau, SAR, PR China. ✉email: wangxuedong@suda.edu.cn; zhengmin@suda.edu.cn; lsliao@suda.edu.cn

In recent years, organic micro/nanomaterials have attracted wide attention as promising building blocks for optoelectronic devices[1–5], owing to their morphology-dependent effect and superior photoelectric characteristics including large optical cross-sections[6,7], ultrafast non-linear responses[8–10], broad wavelength tunability[11,12] and so on[13]. Furthermore, organic micro/nanomaterials inherit the merits of organic semiconductor molecules such as tailorable molecular structure, mechanical flexibility, low-temperature solution processing[14–18]. Compared with single structure, organic heterostructures (OHSs) exhibit unique advantages in integrated optoelectronics, benefiting from their novel performance at heterojunction and multiple functions[19–22]. Among them, one-dimensional (1D) triblock heterostructures with segregated domains and spatial multi-color emissions can manipulate photons and electrons, are particularly attractive micro/nanomaterials for a large variety of applications[23–25]. Besides, owing to the enhanced charge carrier mobility, the organic core/shell heterostructures (OCSSs) are also important for photovoltaic conversion and so on[26–28].

However, the future of organic heterostructures nanotechnology depends on the balance between yield and performance of materials. Therefore, the large-scale synthesis of OHSs is crucial for facilitating their applications in the area of optoelectronics. In last decade, the fabrication of OHSs have been reported by several main approaches, including photochromic[29,30], doping[31–33], heteroepitaxial growth[34–36] and block copolymerization[37–39]. For instance, Prof. Y. W. Zhong et al. reported the synthesis of light-harvesting triblock nanorods via the stepwise co-assembly of polypyridyl Ir(III) and Ru(II) metallophosphors at low acceptor doping ratios. Among them, the organic cocrystal engineering can be used as a promising tool for the fabrication of OHSs owing to their huge material categories and diverse intermolecular interactions[40–42]. A facile approach for the fabrication of OHSs has been previously reported, through which the OHSs are successfully fabricated via the hierarchical self-assembly process from the cocrystals of DPEpe-BrFTA and DPEpe-F$_4$DIB, realizing the precise manipulation for the length ratio of different emissive part in the triblock nanowires by tuning the noncovalent interactions (hydrogen bond and halogen bond)[43]. However, in contrast to single micro/nanostructure, (e.g., nanowires and nanosheets) whose fabrication methods and mechanisms have been intensively investigated, the synthesis of OHSs have largely remained a case-by-case practice, and difficulty construct organic heterostructures in large scale, owing to the occurrence of homogeneous nucleation and phase separation during the co-assembly process of multi-component[44,45]. Particularly, the lattice mismatch of dissimilar cocrystal remains a unsolved problem, which restricting large-scale and fine synthesis of the OHSs. Inspired by alloys (A$_x$B$_{1-x}$) in nature, such as the Al–Mg inorganic alloy, organic alloys as a kind of organic solid with excellent optoelectronic performance, which have been studied in recent years[46,47]. Furthermore, alloys with novel properties tend to have the same crystal lattice as the host material, at the level of a large number of guest addition to the host material.[48] In view of excellent compatibility of the structure, the organic alloy strategy can solve the problem of lattice mismatch and provide the possibility for the large-scale fabrication of OHSs.

In this work, we define a concept of lattice-mismatch-free growth for hierarchical self-assembly of organic semiconductors, which enables the large-scale synthesis of organic heterostructure nanowires. Owing to the organic alloys and cocrystals have similar crystalline structure, we address the problem of lattice mismatch during the self-assembly of organic multicomponent. In detail, the Benzo[ghi]perylene (BGP) selected as the donor, 3,4,5,6-tetrafluorophthalic anhydride (TFPA) and octafluoronaphthalene (OFN) selected as the acceptor, respectively.

Then, the TFPA and OFN molecules leads to the formation of BGP-OFN$_{(0.8)}$-TFPA$_{(0.2)}$ organic alloys, which have desired structural compatibility with BGP-OFN cocrystals. Furthermore, we carefully express the growth mechanism and characterization of two types of OHSs including triblock and core/shell-triblock nanowires based on the lattice-mismatch-free growth of organic semiconductors. In addition, the length ratio of different segments in OHSs can be precisely regulated by finely tuning the stoichiometric ratio of TFPA and OFN molecules. Importantly, various organic triblock nanowires can also be prepared in a large scale following similar procedures. As a proof-of-concept application, we demonstrate the optical applications of the obtained triblock nanowires, including optical logic gates and multicolor photon transmission. This general strategy is of great significance to the large-scale fabrication of OHSs and its application in integrated optoelectronics.

## Results

**Lattice-mismatch-free growth concept from organic cocrystal to alloy.** As shown in Fig. 1a, we firstly prepared BGP-OFN and BGP-TFPA organic cocrystals through solution self-assembly method. Then, the BGP-OFN$_{(0.8)}$-TFPA$_{(0.2)}$ alloy were constructed based on the structurally similar OFN and TFPA molecules. Owing to the excellent structural compatibility of the cocrystal and alloy, two triblock heterostructures were successfully fabricated based on the selective epitaxial growth of the cocrystal and alloy by adjusting the reaction conditions.

The fluorescence microscopy (FM) images of the BGP-OFN cocrystal revealed that the nanowires emit blue light upon excitation by UV light (Fig. 1b). Observed by fluorescence micrograph that the BGP-TFPA nanowires with orange emission under the excitation of UV light (Fig. 1c). In addition, based on the single crystal of BGP-TFPA or BGP-OFN cocrystal, we found that the BGP-TFPA arranged in a sandwich-herringbone motif, while BGP-OFN arranged in an offset face-to-face configuration (Supplementary Figs. 3 and 4). Actually, these two cocrystals are difficult to construct OHSs due to their lattice mismatch (Supplementary Figs. 5 and 6, Supplementary Tables 1–5) leads to the occurrence of phase separation (Supplementary Fig. 7). However, based on the structurally similar OFN (5.498 Å) and TFPA (5.430 Å) molecules, and competitive intermolecular interaction including arene-perfluoroarene (AP) and charge-transfer (CT) interactions, we infer that both of the OFN and TFPA can replace each other, which provides the possibility for the formation of organic alloys[48]. As a proof of the concept, we selected BGP-OFN co-crystal as the host material, and then added a small amount of TFPA molecules to prepare BGP-OFN$_{(1-x)}$-TFPA$_{(x)}$ organic solid solution. In detail, the densities of the TFPA molecules in the host would become greater as x increases. Furthermore, when the assemblies obtained at $x > 20\%$, instead of uniform 1D microwires, sheet-like particles were attached on the microwires, forming branch heterostructures (Supplementary Fig. 8). Therefore, we infer that TFPA can be effectively incorporate into BGP-OFN host at $x = 20\%$. Moreover, we conducted a series of micro-area photoluminescence spectra measurements of multiple organic alloy nanowires. As shown in Supplementary Fig. 9, the emission spectra of multiple alloys from the different area of substrate are almost identical, which further proves that the emission uniformity of organic alloy obtained by this method. Meanwhile, the emission spectra at any positions on BGP-OFN$_{(0.8)}$-TFPA$_{(0.2)}$ alloy are almost identical, which also confirms that the TFPA guest can be uniformly incorporated into the crystal lattice of the BGP-OFN host (Supplementary Fig. 10). Therefore, it can be expected that the 1D nanowires obtained at $x = 20\%$ have BGP-OFN$_{(0.8)}$-TFPA$_{(0.2)}$

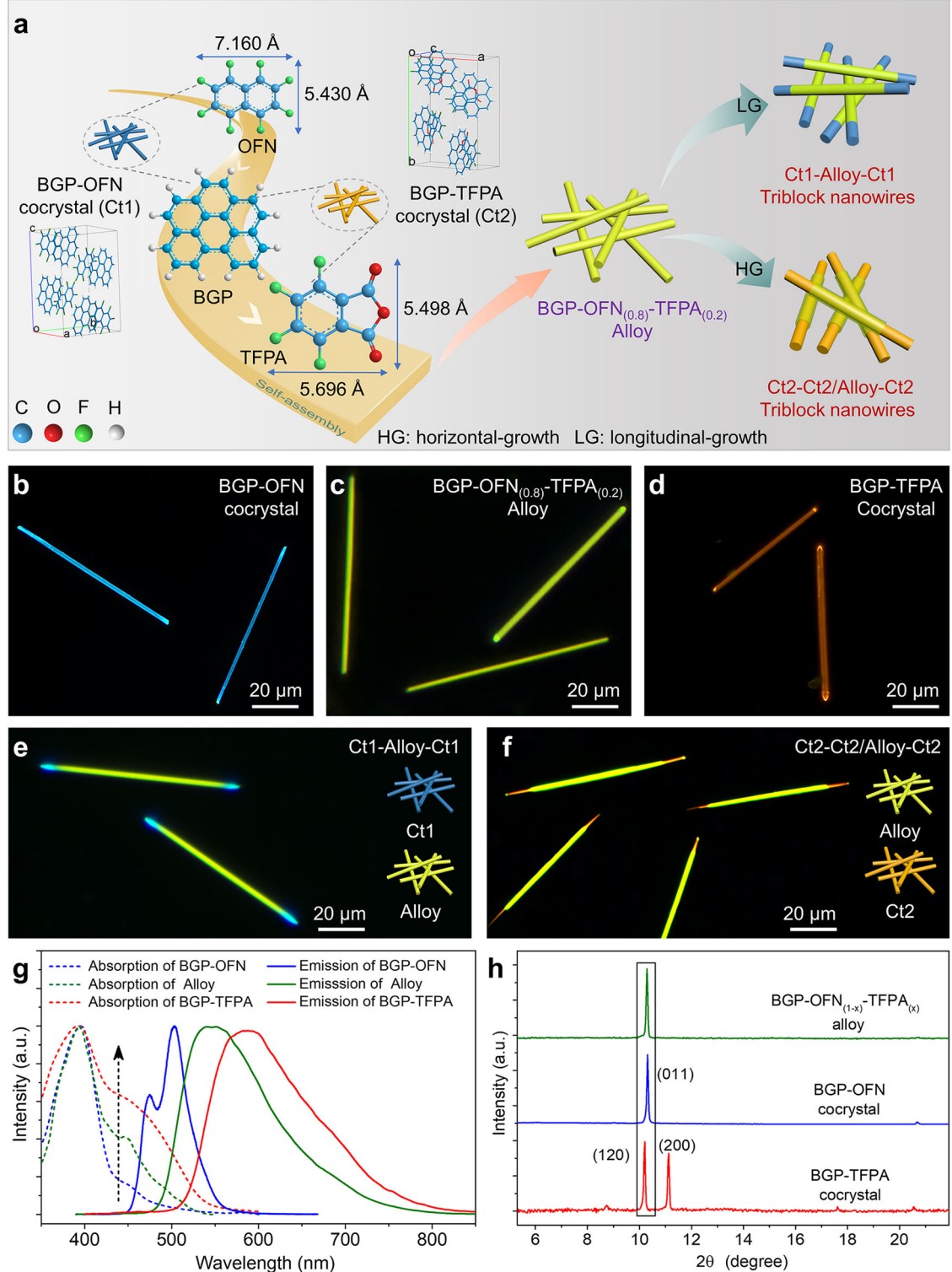

**Fig. 1 Synthesis of two triblock heterostructures via lattice-mismatch-free growth concept. a** Schematic diagram of the co-assembly of cocrystal and alloy into triblock heterostructures. **b–d** FM images of BGP-OFN cocrystals, BGP-TFPA cocrystals and BGP-OFN$_{(0.8)}$-TFPA$_{(0.2)}$ alloys. **e, f** FM images of blue-olive-blue and orange-olive/orange-orange triblock nanowires, respectively. **g, h** The diffuse absorption and PL spectra of BGP-OFN, BGP-TFPA and BGP-OFN$_{(0.8)}$-TFPA$_{(0.2)}$ alloy, and its corresponding XRD pattern.

alloy structures. Figure 1d exhibits the FM images of the BGP-OFN$_{(0.8)}$-TFPA$_{(0.2)}$, the nanowire with the bright olive-emitting under UV light. To obtain more information of the BGP-OFN, BGP-TFPA and BGP-OFN$_{(0.8)}$-TFPA$_{(0.2)}$ alloys, the photoluminescence (PL) and absorption spectroscopy was performed on the three types of crystals. As shown in Fig. 1g, the PL spectrum from

BGP-OFN co-crystals presents two emission peaks at around 473 nm and 500 nm (blue solid curves) due to the AP interaction, whereas, BGP-TFPA co-crystals display a single band at 590 nm (red solid curves) owing to the CT interaction. Moreover, the PL spectrum of the BGP-OFN cocrystal exhibits a large overlap with the absorption spectrum of the BGP-TFPA cocrystal in the

wavelength range of 450–560 nm, which allowing the occurrence of Förster-type energy transfer (FRET)[49]. Therefore, the BGP-OFN$_{(0.8)}$-TFPA$_{(0.2)}$ alloy displays a strong PL single band at 550 nm (green solid curves). In addition, the absorption spectrum of the alloy displays a new absorption band at around 550 nm, which is distinctively different from BGP-OFN (dashed curves, Fig. 1g). This result proves that the BGP-OFN as the host material contains a certain amount of TFPA molecules. In order to further study the crystal structure of the BGP-OFN$_{(0.8)}$-TFPA$_{(0.2)}$ assemblies, X-ray diffraction (XRD) is performed on the BGP-TFPA, BGP-OFN and alloy. As shown in Fig. 1h, the XRD patterns reveal that the alloys are almost isomorphic to BGP-OFN cocrystals. That all, we infer that the BGP-OFN$_{(0.8)}$-TFPA$_{(0.2)}$ alloy is a new crystal species formed by randomly replacing the original OFN molecules with TFPA molecules during the self-assembly process. Furthermore, we also obtained single crystals of BGP-OFN, BGP-OFN$_{(0.8)}$-TFPA$_{(0.2)}$ and BGP-TFPA (Supplementary Fig. 11).

In view of the successfully preparation of BGP-OFN$_{(0.8)}$-TFPA$_{(0.2)}$ alloys and their good lattice matching with BGP-OFN cocrystal, we prepared two triblock heterostructure nanowires. As shown in Fig. 1e, the blue-olive-blue triblock nanowires are shown in the FM images. Notably, based on the different stoichiometric of BGP-OFN cocrystal and BGP-TFPA cocrystal, we can be selectively prepared the blue-olive-blue ($N_{BGP-OFN} \gg N_{BGP-TFPA}$) and orange-olive/orange-orange triblock nanowires ($N_{BGP-OFN} \ll N_{BGP-TFPA}$), in which the orange-olive/orange-orange triblock nanowires composed of BGP-OFN$_{(0.8)}$-TFPA$_{(0.2)}$ alloys and BGP-TFPA cocrystal. As expected, the FM images indicate that the orange-olive-orange triblock nanowires were successfully prepared (Fig. 1f).

**Epitaxial growth of blue-olive-blue triblock nanowires**. The blue-olive-blue triblock nanowires are successfully prepared through epitaxial growth process[36]. In order to investigate the assembly mechanism, we conduct a series of experiments by tuning the time of solvent evaporation, and recording the corresponding FM images of each stage. Herein, we successfully recorded the distinctive four-stage process of growth from nucleation to triblock nanowires, which is schematically represented in Fig. 2a. In the first stage, the BGP-OFN$_{(0.8)}$-TFPA$_{(0.2)}$ alloys firstly nucleates from the mixed solution (DCM, n-hexane, ethanol) and elongates to 1–10 micrometers in three seconds (Fig. 2b) due to the lower solubility of TFPA component and the strong CT interaction with BGP. In the second stage, with the solution evaporates, the alloy nanowires grow to tens of micrometers (Fig. 2c). Subsequently, the TFPA all participate in the self-assembly of the BGP-OFN, the growth of the alloy will be over. In the third stage, the unconsumed BGP and OFN components continue to be assembled along the axial direction of the alloy nanowires, and form a triblock nanowire with two blue-emitting tips (Fig. 2d, third stage). Finally, the BGP-OFN crystal continue grew along on the two tips of the triblock nanowires, until the solvent is completely evaporated (Fig. 2e). In addition, we also collected the emission spectra of the middle part of the triblock nanowire at different growth stages. The results show that the emission of the triblock microwires obtained in each step is almost identical, the middle part corresponds to the BGP-OFN$_{(0.8)}$–TFPA$_{(0.2)}$ alloy, and the two tips correspond to BGP-OFN cocrystal (Supplementary Fig. 12). In order to study the structural of the triblock nanowires, we further characterized the photoluminescence of a single nanowire. As shown in Fig. 2f, under bright field, the entire nanowire shows uniform single color without obvious interface owing to the good lattice matching of BGP-OFN and alloy[43]. Moreover, the FM images of the single

nanowire indicate that the two tips of the nanowire emit blue light, while the central segment emits olive light, under the excitation UV light (Fig. 2g). In addition, Fig. 2h shows that the blue-emitting tip (1) with two peaks of PL at around 475 and 500 nm, which corresponds to the BGP-OFN emission. The olive segment (3) shows a broad emission spectrum with single peak at 550 nm corresponding to the BGP-OFN$_{(0.8)}$-TFPA$_{(0.2)}$ alloy emission. Meanwhile, the emission spectra at different positions of the middle part are also almost identical, which further proves the uniformity of alloy components (Supplementary Fig. 13). Furthermore, Transmission electron microscope (TEM) and its corresponding selected area electron diffraction (SEAD) patter demonstrate that these nanowires are high-quality single crystals with uniform size (Fig. 2i). The amplified SAED inset in Fig. 2i suggests that both the central part and end of the triblock nanowires growing along the [100] direction due to the similar structure of BGP-OFN cocrystals and their alloys.

**Large-Scale synthesis of blue-olive-blue triblock nanowires**. According to the structurally similar organic alloy and cocrystal (Fig. 1h), we define a concept of lattice-mismatch-free for large-scale synthesis of heterostructures, which avoid the lattice mismatch problem during the hierarchical self-assembly of organic semiconductor molecules. As a typical example, we carefully recorded the fluorescence microscopy (FM) images of triblock nanowires in different areas of the quartz substrate (Fig. 3a). At the same time, we also performed the corresponding statistical analysis to demonstrate our claims of "large scale" synthesis. Figures 3b–e shows the typical FM images of blue-olive-blue triblock nanowires, respectively. It can be seen that a large number of triblock nanowires with the length of 40–50 μm are distributed in different areas of substrate (Fig. 3f). At the same time, other areas also have a large number of blue-olive-blue triblock nanowires, which proved that the triblock nanowires were successfully synthesized in large scale. This robust strategy paves the avenue to the fine synthesis of heterostructures in a large scale and facilitate their applications in organic optoelectronics at micro/nanoscale.

**Epitaxial growth of core/shell triblock nanowires**. Based on the BGP-TFPA cocrystal and BGP-OFN$_{(0.8)}$-TFPA$_{(0.2)}$ alloy, we also propose a facile approach for the fabrication of core/shell triblock nanowires in large scale. Figure 4a schematically depicts the horizontal epitaxial growth of BGP-OFN$_{(0.8)}$-TFPA$_{(0.2)}$ alloy on the surface of BGP-TFPA nanowires, in which the thickness of shell can be regulated by tuning the experimental temperature. Supplementary Fig. 14 shown the bright field images of the self-assembly process of a typical orange-olive/orange-organic triblock nanowires captured at different growth times. It can be seen that, in the first stage, the BGP-TFPA cocrystal as core nanowire firstly nucleates and growth from the mixed solution due to the lower solubility of TFPA component and the strong CT interaction with BGP. Then, with the solution evaporates, the shell covers the whole core wire in the beginning and during the shell and core growth. Finally, the unconsumed BGP and TFPA components continue to assemble into core nanowire and elongates along the longitudinal direction until the solvent is completely evaporated. Notably, the slight difference in the XRD patterns of the alloy and the BGP-TFPA cocrystal indicates that the low lattice mismatching between alloy and BGP-TFPA ($f = 1.0 \%$), which allow the occurrence of epitaxial growth process of the core/shell triblock nanowires (Supplementary Fig. 15). In addition, the core-shell nanowire is not completely covered by the alloy, so it can be regarded as another triblock nanowire. At the same time, we can obtain two types of core/shell triblock

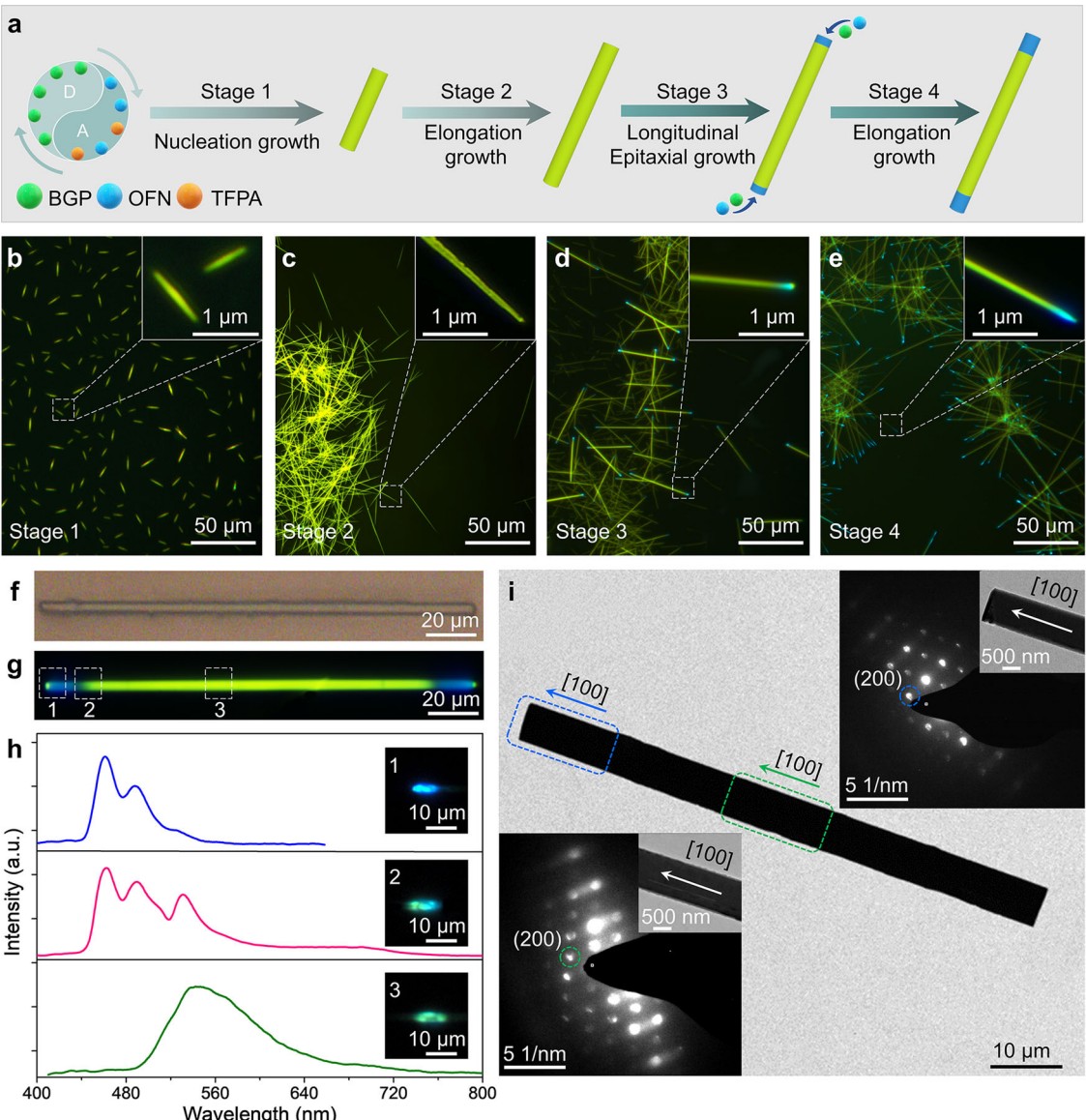

**Fig. 2 The growth mechanism and characterization of the blue-olive-blue triblock heterostructure nanowires. a** Schematic representation of the four-stage growth process of blue-olive-blue triblock heterostructures. FM images recorded during stage 1 (**b**), 2 (**c**), and 3 (**d**), 4 (**e**), respectively. The scale bar is 50 μm. The growing tips are magnified in the white squares. **f, g** Bright-field and FM image of individual blue-olive-blue triblock nanowire. The scale bar is 20 μm. **h** PL spectra collected from different sections marked in (**g**). The inset corresponds to the FM images of different positions excited by laser beam $\lambda = 375$ nm. The scale bar is 10 μm. **i** TEM images of triblock nanowires. The scale bar is 10 μm. Insets: corresponding SAED patterns of the central part (lower left) and the tip (upper right) of a single triblock nanowire.

nanowires due to the different thickness of the shell (Figs. 4b, c). As shown in Fig. 4c, the core/shell triblock nanowire obtained has a smaller shell thickness due to the faster evaporation of mixed solvent at 25 °C. In contrast, the core/shell triblock nanowire obtained has a bigger shell thickness due to the slower evaporation of mixed solvent at 15 °C (Fig. 4b). It can be clearly seen that as the thickness increases, the emission color of the middle part from the orange to olive. As shown in Fig. 4e, f, the scanning electron microscope (SEM) results further confirm that the microstructure of nanowires with different thickness of shell layer. To in-depth understand the microstructure of triblock nanowires, the corresponding PL spectra were performed. As shown in Fig. 4d, the spatially resolved spectra at different positions of the core-shell nanowires show that the emission of the core is consistent with the emission spectrum of the BGP-TFPA eutectic, proving that the core-shell structure is composed of the BGP-TFPA as the core and

the alloy as the shell. It is worth noting that the PL spectra of the core and the shell segments mostly overlap, and the emission peak difference is only 50 nm, which leading to the two-fluorescence emission coupled into one emission peak (560 nm). In addition, the center part of the nanowires (core/shell marked in Fig. 4b) has consistent emission with junction, which confirms that the center part is composed of core and shell. The inset respectively corresponds to the confocal laser scanning microscopy image of single nanowire at different positions (Fig. 4b) excited with laser beam ($\lambda = 375$ nm). In order to obtain the detailed information on the crystal structure of core/shell triblock nanowires, we also investigated the XRD patterns (Supplementary Fig. 15) of BGP-TFPA and core/shell triblock nanowires, etc. Obviously, the main peaks of core/shell structures belong to the outer shell composed of alloy, whereas some weak peaks, such as 10.06° and 11.10° derive from core composed of BGP-TFPA.

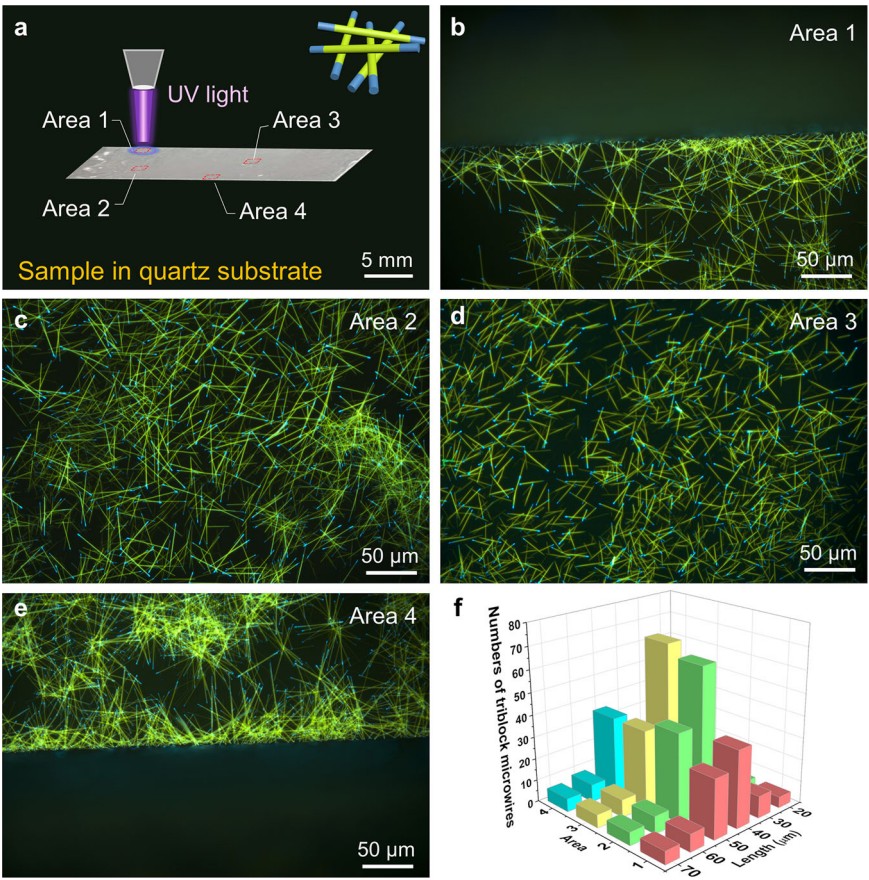

**Fig. 3 Large-scale synthesis of blue-olive-blue triblock nanowire. a** Randomly selecting the area to characterize the distribution of the blue olive blue triblock heterostructure in the quartz substrate. FM images of blue-olive-blue triblock nanowires in the area 1 (**b**), area 2 (**c**), area 3 (**d**) and area 4 (**e**). **f** The histogram of length distribution of triblock heterostructures, by randomly measuring the FM images of a large number of microwire samples. The length of the triblock nanowires ranges from 40 to 50 μm.

**Controlled synthesis of organic heterostructure nanowires.** To date, the precisely controlled synthesis of triblock nanowires is rarely considered because of the difficult control on the size and shape of each organic components as well as the selective nucleation and growth of the secondary organic components. In this work, the length ratio ($\alpha$) of the multicolor-emissive part in the triblock nanowire (Fig. 5a, f) is successfully controlled by carefully adjusting the stoichiometric ratio of OFN and TFPA components ($n_{TFPA}/n_{OFN}$). Clearly, the corresponding FM images of a series of triblock heterostructures show that the length ratio of the alloys at center in the triblock nanowire is negatively correlated with the ratio between the OFN and the TFPA materials as verified in Fig. 5b–e. In addition, the lengths of the two blue parts of the triblock nanowires are both about 5.5 μm (Fig. 5c), which indicates that the blue-olive-blue triblock nanowires has a high degree of structural symmetry. Likewise, the controlled synthesis of core/shell triblock nanowires is shown in Figs. 5g–j. Moreover, Fig. 5h also suggests that the orange-olive/orange-orange triblock nanowires has a high structural symmetry. Moreover, the wrapped degree of the core at center part in the core/shell nanowires can be effectively controlled via adjusting the amount of TFPA under the experiment temperature at 25 °C. In addition, we have carefully discussed the crystal distribution and characterized their length and/or spectral emission uniformity in different molar ratios. As shown in Supplementary Figs. 16 and 17, the triblock nanowires exhibit a certain length uniformity in the same area of the quartz substrate. At the same time, the emission spectra of the middle part of triblock nanowires in the different molar ratios are also almost identical, which further

proves that the emission uniformity of triblock nanowires obtained by this method (Supplementary Fig. 18).

**General synthesis of organic heterostructure nanowires.** In view of the large-scale synthesis of blue-olive-blue triblock heterostructure composed of BGP-OFN cocrystal and BGP-OFN$_{(0.8)}$-TFPA$_{(0.2)}$ alloy, we further proposed a general strategy for the large-scale synthesis of triblock heterostructure nanowires (Fig. 6a). Firstly, we selected three structurally similar molecules including TFPA, TCPA and TBPA as acceptor to construct organic alloy, namely BGP-OFN$_{(0.8)}$-TCPA$_{(0.2)}$ and BGP-OFN$_{(0.8)}$-TBPA$_{(0.2)}$. Owing to the organic alloy and cocrystal have similar crystal structure, we successful realized the lattice-mismatch-free growth of three organic heterostructure nanowires in large scale. Notably, the large-scale synthesis of type 1 triblock nanowires (Fig. 6b and Supplementary Fig. 19), namely core/shell triblock nanowires composed of BGP-TFPA cocrystal and BGP-OFN$_{(0.8)}$-TFPA$_{(0.2)}$, their reaction conditions must be carefully fine-tuned to avoid homogeneous nucleation and phase separation of the various components in the reagents. Compared with type 1 triblock nanowires, the type 2 and type 3 triblock nanowires can be readily prepared owing to the BGP-OFN and their alloy have similar crystal structure (Fig. 6c, d). Notably, the spectrum at the junction includes the two emissions (Supplementary Figs. 33 and 34, Supplementary Note 2), which implies that the multicolor emission characteristics of these heterostructures make them possible to be applied for the optical waveguides and optical logic gate operation (Supplementary Figs. 20–26, Supplementary Note 1).

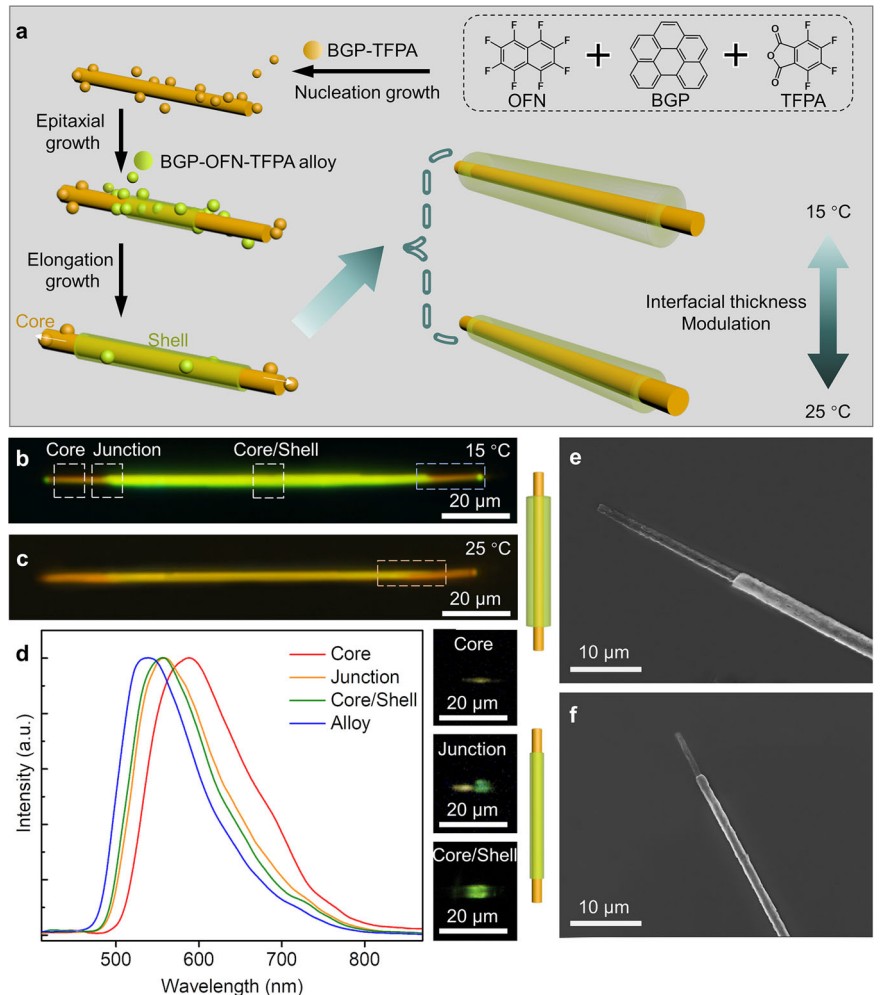

**Fig. 4 The synthesis and characterization of core/shell triblock nanowires. a** Schematic illustration of the synthesis of orange-orange/olive-orange triblock nanowires. **b** FM images of triblock nanowires formed at 15 °C. **c** FM images of triblock nanowires formed at 25 °C. **d** PL spectra collected from three sections marked in (**b**). Insets are the corresponding FM images excited by laser beam $\lambda = 375$ nm. **e**, **f** SEM images of two types of triblock nanowires with different thickness.

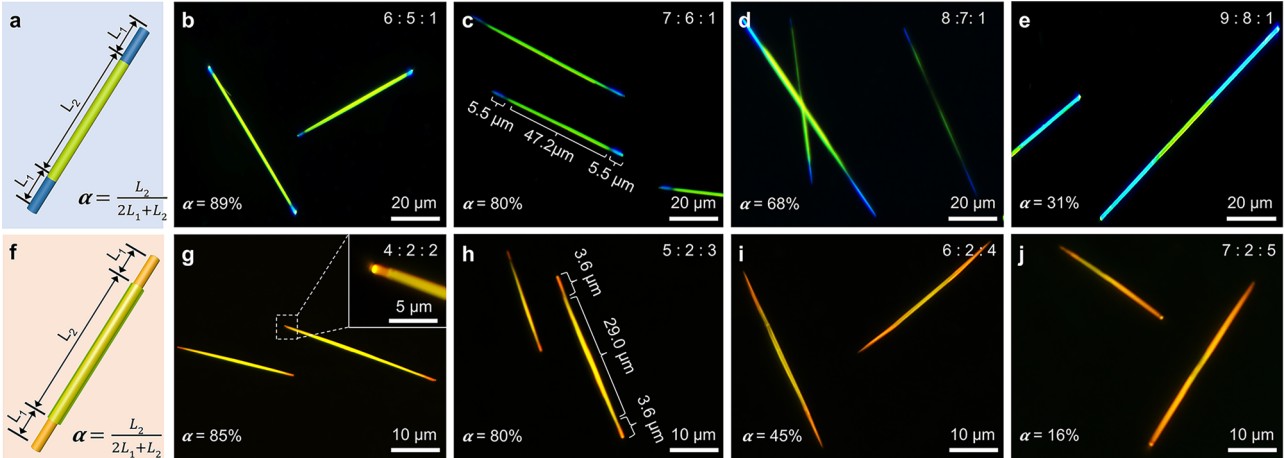

**Fig. 5 Elaborate control for the length ratio of the two segments of triblock heterostructure nanowires. a** illustration for the length ratio between the blue part and the olive part of the triblock nanowire. **b–e** FM images of the blue-olive-blue triblock nanowires with different length ratio, the molar ratio of BGP, OFN and TFPA was set to 6:5:1 (**b**), 7:6:1 (**c**), 8:7:1 (**d**), 9:8:1 (**e**). **f** Schematic illustration of the length ratio of the orange segment to the olive segment of the triblock nanowire. **g–j** FM images of the orange-orange/olive-orange triblock nanowires with different length ratio, the molar ratio of BGP, OFN and TFPA was set to 4:2:2 (**g**), 5:2:3 (**h**), 6:2:4 (**i**), 7:2:5 (**j**).

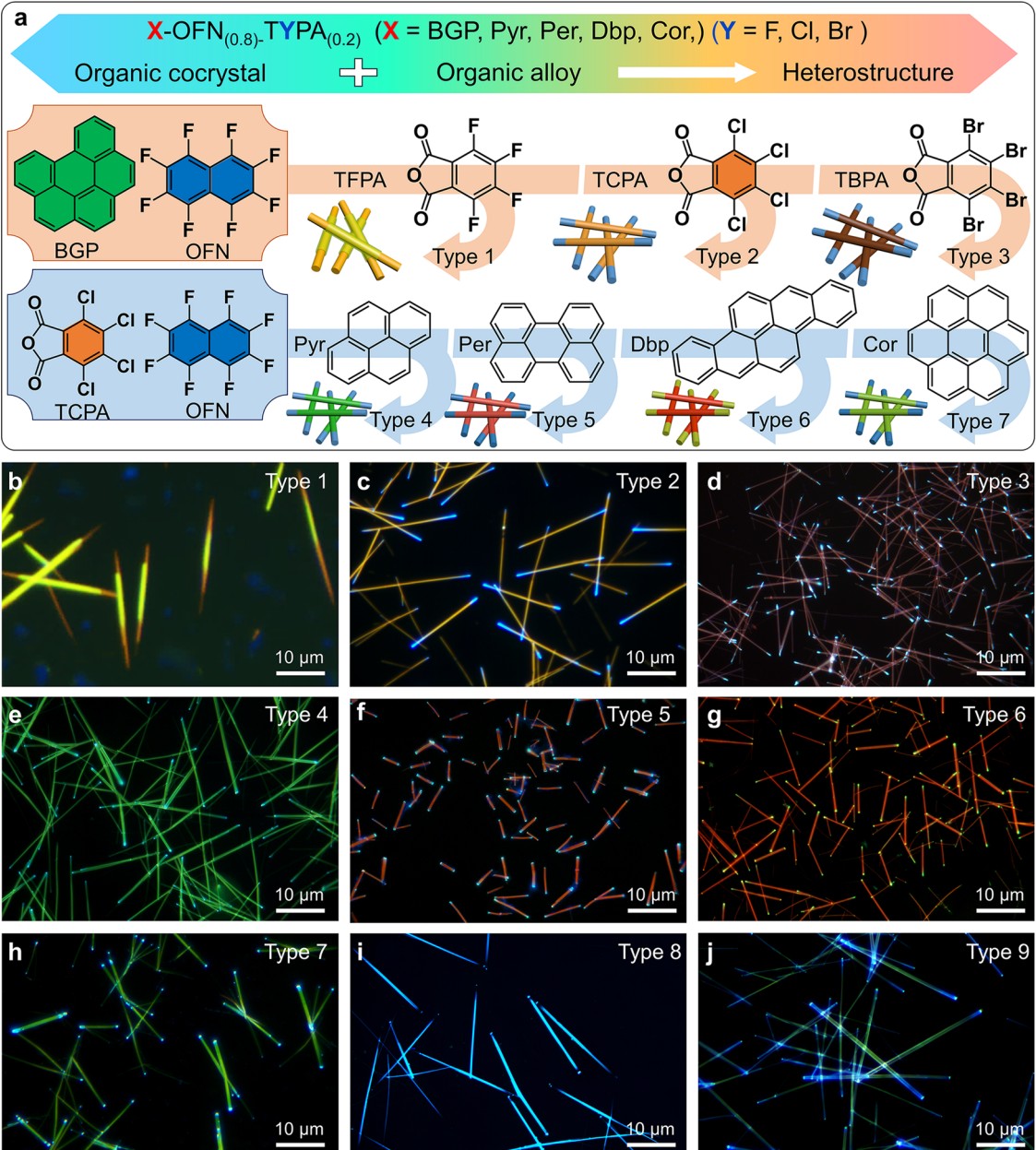

**Fig. 6 Synthesis of nine types of triblock nanowires. a** Scheme for the synthesis of the nine types of triblock nanowires composed of organic X-TYPA (X = BGP, Pyr, Per, Dbp, Cor, Trp, Dba) (Y = F, Cl, Br) cocrystals and the corresponding X-OFN$_{(0.8)}$-TYPA$_{(0.2)}$ alloys. **b–j** FM images of heterostructure nanowires under UV excitation. All scale bar is 10 μm.

Based on these acceptors, we further selected a number of donors (Pyr, Per, Dbp, Cor et al.) to achieve a series of triblock nanowires (Fig. 6a and Supplementary Fig. 27). As shown in Fig. 6e–j, the FM images of various nanowires confirmed that we the have been successfully prepared a series of organic cocrystal (Supplementary Fig. 30) and corresponding heterostructure nanowires through different donor and acceptor molecules. The PL spectra indicate that this heterostructure nanowires were composed of cocrystals and alloys (Supplementary Figs. 28, 29 and 31). As shown in Supplementary Fig. 32, the FM images of various nanowires further verified that we have been successfully prepared heterostructure nanowires in large scale. This founding makes the bottom-up approach competitive with traditional top-down methods in the large-scale synthesis of complex hierarchical nanowires, thereby providing opportunities for construction of various organic heterostructure nanowires with superior performances.

## Discussion

In Summary, we have successfully fabricated various organic triblock nanowires in a large scale via the lattice-mismatch-free growth process from organic alloy to cocrystal. Notably, this strategy readily allows the construction of organic heterostructures through solution self-assembly methods. At the same time, the length ratio of different segments of the triblock nanowires can be precisely regulated by changing the stoichiometric ratio of different components. Owing to the triblock nanowires have multicolor emission and multichannel coupling output characteristics, which are used as an optical logic gate application platform to realize advanced photonic signal processing. Our strategy provides a insight for the large-scale preparation of heterostructure nanowires, which lays the materials foundation for integrated optoelectronics in future.

## Methods

**Materials.** Benzo[ghi]perylene (BGP, CAS: 191-24-2), Pyrene (Pyr, CAS: 129-00-0), Perylene (Per, CAS: 198-55-0), Dibenzo[b,def]chrysene (Dbp, CAS: 189-64-0), Coronene (Cor, CAS: 191-07-1), Dibenz[a,h]anthracene (Dba, CAS: 53-70-3), Triphenylene (Trp, CAS: 217-59-4), 3,4,5,6-tetrafluorophthalic anhydride (TFPA, CAS: 652-12-0), Tetrabromophthalic anhydride (TBPA, CAS: 632-79-1), Tetrachlorophthalic anhydride (TCPA, CAS: 117-08-8), Octafluoronaphthalene (OFN, CAS: 313-72-4) were purchased from Sigma-Aldrich Co. The dichloromethane ($CH_2Cl_2$, A.R.), methanol (A.R.) and ethanol (A.R.), cyclohexane (A.R.) and n-hexane (A.R.) solvents, were purchased from Beijing Chemical Ltd. China. In addition, all compounds and solvents were used without further treatment.

**Self-assembly method of organic BGP-OFN and BGP-TFPA cocrystals.** In a typical experiment, 0.1 mmol (27.63 mg) BGP and 0.1 mmol (27.21 mg) OFN was dissolved in 10 mL dichloromethane (DCM), in which the mixture of BGP and OFN with 1:1 molar ratio. Then this DCM solution include BGP and OFN components was added into 20 mL of ethanol, which means that volume ratio of DCM and ethanol is 1:2. Then the mixed solution was directly drooped onto the quartz substrate at room temperature in the air. After evaporating the solvent, BGP-OFN cocrystals with blue color were achieved. Similarly, 0.1 mmol (27.63 mg) BGP and 0.1 mmol (22.01 mg) TFPA was dissolved in 10 mL DCM, in which the components of BGP and TFPA with 1:1 molar ratio. Then this DCM solution include BGP and TFPA components was added into 20 mL of n-hexane, which means that volume ratio of DCM and n-hexane was 1:2. Then they were directly dropped onto the quartz substrate at room temperature in the air, and the BGP-TFPA cocrystals with orange color were obtained after the solvent evaporated totally.

**Self-assembly method of organic BGP-OFN$_{(0.8)}$-TFPA$_{(0.2)}$ alloy.** Typically, 10 mL of the monomer solutions containing 0.1 mmol (27.63 mg) BGP, 0.08 mmol (21.76 mg) OFN and 0.02 mmol (4.4 mg) TFPA in DCM ($N_{OFN}$ / ($N_{OFN}$ + $N_{TFPA}$) = 0.8, $N_{BGP}$ = $N_{OFN}$ + $N_{TFPA}$, $C_{BGP}$ = 10 mM) were quickly injected into 20 mL of ethanol solvent ($V_{DCM}$ / $V_{ethanol}$ = 1:2). Then they were directly dropped onto the quartz substrate at room temperature in the air, and the BGP-OFN$_{(0.8)}$-TFPA$_{(0.2)}$ cocrystals with olive color were obtained after the solvent evaporated totally.

**Hierarchical self-assembly method of blue-olive-blue triblock nanowires.** In a typical experiment, 0.06 mmol (16.58 mg) BGP, 0.05 mmol (13.60 mg) OFN and 0.01 mmol (2.2 mg) TFPA ($N_{BGP}$ = $N_{OFN}$ + $N_{TFPA}$, $N_{OFN}$ / ($N_{OFN}$ + $N_{TFPA}$) = 0.83) were dissolved in 10 mL DCM solvent. Next, the stock solution was quickly injected into the mixed solvent including 14 mL n-hexane and 7 mL ethanol ($V_{n-hexane}$ / $V_{ethanol}$ = 1:2) ($V_{DCM}$ / $V_{Mix}$ = 1:2). The dispersed solution was dropped onto a quartz substrate, which was then covered with a watch glass at room temperature in the air. The triblock heterostructures were observed after the solvent completely evaporated. The type 2 and type 3 triblock nanowires were prepared following similar solution-based preparation procedures to blue-olive-blue triblock nanowires (Supplementary Fig. 1).

**Hierarchical self-assembly method of orange-olive/orange-orange core/shell triblock nanowires.** In a typical experiment, 0.05 mmol (13.82 mg) BGP, 0.02 mmol (5.44 mg) OFN and 0.03 mmol (6.60 mg) TFPA ($N_{BGP}$ = $N_{OFN}$ + $N_{TFPA}$, $N_{TFPA}$/($N_{OFN}$ + $N_{TFPA}$) = 0.6) were dissolved in 10 mL DCM solvent at 25 °C. Next, the stock solution was quickly injected into mixed solvent including 15 mL cyclohexane and 5 mL methanol. Then, the dispersed solution was dropped onto a quartz substrate at 25 °C in the air. The core/shell triblock heterostructures with a small shell thickness were obtained after the solvent completely evaporated. The orange-olive/orange-orange core/shell triblock nanowires with a big shell thickness were also synthesized following the similar experiment procedure, except the reaction temperature was set to 15 °C (Supplementary Fig. 2).

## Data availability

All data generated in this study are provided in the paper or Supplementary Information. Additional data related to this paper may be requested from the authors.

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

## Acknowledgements

This work was supported by the National Natural Science Foundation of China (Grant Nos. 52173177, 21971185, X.W.). This project is also funded by the Collaborative Innovation Center of Suzhou Nano Science and Technology (CIC-Nano), and by the "111" Project of the State Administration of Foreign Experts Affairs of China.

## Author contributions

X.-D. W., M. Z. and L.-S. L. designed the experiments; Q. L. synthesized the organic heterostructure nanowires. Q. L. and Y. Y. performed the structural/optical characterizations. Y. Y. performed the optical waveguide properties. Q. L. performed the simulation of the predicted morphologies of organic molecules. Q. L., X.-D. W., M.-P. Z. and L.-S. L. discussed the interpretation of results and wrote the paper. All authors discussed the results and commented on the manuscript.

## Competing interests

The authors declare no competing interests.
