## [Peer Review File · Nature Communications]

Lattice-Mismatch-Free Growth of Organic Heterostructure Nanowires from Cocrystals to AlloysReviewers' comments:

Reviewer #1 (Remarks to the Author):

The manuscript by Lv et. al., discusses the large-scale synthesis of organic heterostructures. They have used a lattice-mismatch-free growth method to accomplish this. This article is addressing an important problem in organic heterostructures synthesis and the work is suitable for publication in Nature Communications after addressing the below raised concerns.

- i) On page 6 it says that "it can be expected that the 1D nanowires obtained at $x = 20\%$ have BGP-OFN(0.8)-TFPA(0.2) alloy structures" However there are no experimental evidences supporting that the alloys contain the same mole ratio. Generally, NMR is used to determine the mole ratio of alloys/co-crystals. Authors can dissolve wires in CDCl_3 and record their NMR to determine the mole percentage of OFN and TFPA in nanowires.
- ii) It is not clear to me why in the case of orange-olive/orange-organic triblock nanowires the shell is not covering the entire nanowire. Does the shell cover the whole wire in the beginning and during the shell growth, the wire elongates along the longitudinal direction? Time dependent studies should be done to address this.
- iii) Figure 4a is confusing. It shows the shell structure forms separately, which then covers the nanowire. I think this is misleading, authors should consider redepicting the process in Figure 4a.

Minor corrections:

- iv) On page 6 authors have used abbreviations "AP and CT" but did not mention the full form anywhere in the draft
- v) On page 3 there is a typo. Last few lines read DPEpe-BrFTA and DPEpe-BrFTA

Reviewer #2 (Remarks to the Author):

Manuscript NCOMMS-21-44837-T reports on the synthesis of organic wires based on charge transfer crystals of BGP with other acceptor molecules. By a careful choice of acceptor molecules the authors are able to demonstrate that cocrystal alloys can be used as a way to grow heterostructures while retaining crystalline structures at the junction. While this is in principle interesting, I do not think that this paper merits the audience of Nature Communication. In my opinion the authors do not show that this is general as they write in the title, since this appears to work for a very specific set of molecules and for very specific crystal shape which is the one of nanowires. The optical characterization and logic gate experiments have been already reported by the same authors and other in previous publications (for example ref. 43, 44).

Before resubmission to a more specialized journal, I urge the authors to revise the manuscript and try to have a more quantitative analysis of the optical data, for example by fitting the spectra with gaussian bands and report spectral shifts or spectral weights in a more quantitative and consistent way. In the present version often the authors write about colors (blue, orange, olive sometimes called yellow). This is confusing and leaves space for interpretation.

RE: Revision for Manuscript ID NCOMMS-21-44837A-Z

Answers to Reviewer #1

General comment: The manuscript by Lv et. al., discusses the large-scale synthesis of organic heterostructures. They have used a lattice-mismatch-free growth method to accomplish this. This article is addressing an important problem in organic heterostructures synthesis and the work is suitable for publication in *Nature Communications* after addressing the below raised concerns.

Response: We sincerely thank the reviewer for the kind evaluation on our work. We have carefully answered reviewer's comments and provided corresponding explanations in the revised manuscript. The detailed point-by-point responses of the reviewer's comments are as follows. We believe that the revised version is suitable for the publication in *Nature Communication*.

i) On page 6 it says that "it can be expected that the 1D nanowires obtained at $x = 20\%$ have BGP-OFN_(0.8)-TFPA_(0.2) alloy structures" However there are no experimental evidences supporting that the alloys contain the same mole ratio. Generally, NMR is used to determine the mole ratio of alloys/co-crystals. Authors can dissolve wires in CDCl₃ and record their NMR to determine the mole percentage of OFN and TFPA in nanowires.

Response: We greatly appreciate reviewer's kind suggestion to help us strengthen the manuscript. We tried to carry out a series of NRM characterizations of BGP-OFN_(0.8)-TFPA_(0.2) organic solid solutions obtained by collecting the assemble organic microcrystals on a quartz substrate (Figure R1a). Owing to the absence of hydrogen atom of these acceptor molecules such as OFN, TFP, TFPA (Figure R1b), we have to perform the quantitative ¹³C-NMR instead of ¹H-NMR measurement to evaluate the composition ratio. Unfortunately, we are sorry about that we not obtain clear C=C signals in quantitative ¹³C-NMR spectrum of three molecules to analyze the composition ratio of organic alloys (Figure R2), owing to the low solubility of these

organic molecules in CDCl_3 or DMSO-d_6 (Figure R1c). It can be seen that a small amount of alloy powders (8 mg) has been supersaturated and precipitated into crystals in 0.5 mL CDCl_3 at room temperature, which make it difficult to obtain the high quality ^{13}C -NMR signals. However, photograph and FM (Figure R1a, Figure R3a-c) observation results reveal that the organic alloy nanowires over large area of substrate display uniform and emission olive light. Thereby, we believe that the acceptor molecules were completely and uniformly added into the BGP-OFN host in the preparation ratio. As a proof of concept, we conducted a series of micro-area photoluminescence spectra measurements of multiple organic alloy nanowires to verify our claim. As shown in Figure R3d, the emission spectra of multiple alloys from the different area of substrate are also almost identical, which further proves that the emission uniformity of organic alloy obtained by this method. Meanwhile, the emission spectra at any positions on BGP-OFN_(0.8)-TFPA_(0.2) alloy are almost identical, which also confirms that the TFPA guest can be uniformly incorporated into the crystal lattice of the BGP-OFN host (Supporting Information, Figure S10). Thus, we think that the composition ratio of organic solid solution can be equivalently regarded as the preparation ratio.

Revision: On Page 6, line 13-20, some sentences have been added and revised in the manuscript. All the changes have been highlighted in the revised manuscript. The Figure R3 has been added in the revised supporting information.

Figure R1. (a) Photograph of assemble organic solid solutions on a quartz substrate ($7.5\text{ cm} \times 2.5\text{ cm}$) taken under a UV lamp (330–380 nm) (b) The molecular structures of acceptors and donors. (c) Magnified images of NMR tubes with BGP-OFN-TFP organic solid solutions in CDCl_3 .

Figure R2. ^{13}C -NMR spectra of BGP, OFN, TFPA and their BGP-OFN_(0.8)-TFPA_(0.2) organic alloys, and the corresponding parameters of NMR measurement.

Figure R3. (a-c) Fluorescence microscopy images of organic alloy nanowires. (b) The corresponding spatially resolved PL spectra of obtained BGP-OFN_(0.8)-TFPA_(0.2) nanowires on different area of substrate.

ii) It is not clear to me why in the case of orange-olive/orange-organic triblock nanowires the shell is not covering the entire nanowire. Does the shell cover the whole wire in the beginning and during the shell growth, the wire elongates along the longitudinal direction? Time dependent studies should be done to address this.

Response: We greatly appreciate the reviewer's comments to help us strengthen our work. We have done time dependent studies of orange-olive/orange-organic triblock nanowires to clarify the growth process of core/shell nanowires (Figure R4). It can be

seen that, in the first stage, the BGP-TFPA cocrystal as core nanowire firstly nucleates and growth from the mixed solution due to the lower solubility of TFPA component and the strong CT interaction with BGP. Then, with the solution evaporates, the shell covers the whole core wire in the beginning and during the shell and core growth. Finally, the unconsumed BGP and TFPA components continue to assemble into core nanowire and elongates along the longitudinal direction until the solvent is completely evaporated.

Revision: On Page 11, line 3-4, some sentences have been added and revised in the manuscript. “Figure S14 shown the bright field images of the self-assembly process of a typical orange-olive/orange-organic triblock nanowires captured at different growth times.” All the changes have been highlighted in the revised manuscript. The Figure R4 has been added in the revised supporting information.

Figure R4. (a-k) The self-assembled growth process of a typical orange-olive/orange-organic triblock nanowire recorded by a bright-field microscope with a scale bar of 20 μm. (l) The fluorescence microscopy image of corresponding as-prepared orange-olive/orange-organic triblock nanowires. (m) The time-dependent length of the whole core/shell triblock nanowire.

iii) Figure 4a is confusing. It shows the shell structure forms separately, which then covers the nanowire. I think this is misleading, authors should consider rededicating the process in Figure 4a.

Response: Thanks for the reviewer's valuable comments. We greatly appreciate reviewer's kind suggestion to help us strengthen the manuscript. We have rededicated the schematic illustration of the synthesis of orange-orange/olive-orange triblock nanowires in Figure R5.

Revision: All the changes have been highlighted in the revised manuscript. The Figure R5 has been replaced the Figure 4 in the revised manuscript.

Figure R5. The synthesis and characterization of core/shell triblock nanowires. (a) Schematic illustration of the synthesis of orange-orange/olive-orange triblock nanowires. (b) FM images of triblock nanowires formed at 15°C. (c) FM images of triblock nanowires formed at 25°C. (d) PL spectra collected from three sections marked in (b). Insets are the corresponding FM images excited by laser beam $\lambda = 375$ nm. (e, f) SEM images of two types of triblock nanowires with different thickness.

iv) On page 6 authors have used abbreviations “AP and CT” but did not mention the full form anywhere in the draft.

Response: Thanks for the reviewer's careful examines and valuable comments. We have added the full form for the “AP and CT” in the corresponding statement.

Revision: On Page 6, line 4-5, some sentences have been added and revised in the manuscript. “competitive intermolecular interaction including arene-perfluoroarene (AP) and charge-transfer (CT) interactions” All the changes have been highlighted in the revised manuscript.

v) On page 3 there is a typo. Last few lines read DPEpe-BrFTA and DPEpe-BrFTA

Response: Thanks for reviewer's comments and feedback. We have revised the corresponding statement.

Revision: On page 3, line 19, some sentences have been revised in the manuscript. “through which the OHSs are successfully fabricated via the hierarchical self-assembly process from the cocrystals of DPEpe-BrFTA and DPEpe-F₄DIB.” All the changes have been highlighted in the revised manuscript.

Answers to Reviewer #2

General comment: Manuscript NCOMMS-21-44837-T reports on the synthesis of organic wires based on charge transfer crystals of BGP with other acceptor molecules. By a careful choice of acceptor molecules the authors are able to demonstrate that cocrystal alloys can be used as a way to grow heterostructures while retaining crystalline structures at the junction.

Response: Thanks for reviewer's time and efforts in examining the manuscript. We greatly appreciate the reviewer's critical comments to help us strengthen our work. We have added more detailed information in the results and discussion part to demonstrate our claims of "a general large scale synthetic strategy based on the concept of lattice-mismatch-free growth of heterostructure nanowires", including more heterostructure nanowires from different donors and the novel optical applications. At the same time, we carefully revised the errors in the manuscript and supporting information. We hope reviewer can reconsider our manuscript.

Critical comment: While this is in principle interesting, I do not think that this paper merits the audience of Nature Communication. In my opinion the authors do not show that this is general as they write in the title, since this appears to work for a very specific set of molecules and for very specific crystal shape which is the one of nanowires. The optical characterization and logic gate experiments have been already reported by the same authors and other in previous publications (for example ref. 43, 44).

Response: So that, the Reviewer #2 indicates it is not suitable to accept this work in *Nature Communication*, this is a crucial concern so we respond it in three aspects:

1) very specific set of molecules?

Response: In the previous version, we are sorry about that we not provide more example to demonstrate our claim "general synthesis of organic heterostructure nanowires", which lead to reviewer think that this strategy appears to a very specific set of molecules. Herein, we have added more example to verify that this general

strategy in the present version. Actually, based on the concept of lattice-mismatch-free growth for hierarchical self-assembly of organic semiconductors, we have achieved a library of tens distinct organic heterostructure nanowires through some of the molecules currently available in our laboratory. As shown in Figure R6 and R7, the FM images of various nanowires confirms that we the have been successfully prepared a series of organic cocrystals and corresponding heterostructure nanowires through different donor and acceptor molecules. The PL spectra indicate that this heterostructure nanowires were composed of cocrystals and alloys (Figure R8). At the same time, the FM images displays synthesis of various heterostructures nanowires in a large scale (Figure R9). This founding makes the bottom-up approach competitive with traditional top-down methods in the large-scale synthesis of complex hierarchical nanowires, and thus provides a new opportunity for construction of various organic heterostructure nanowires with superior performances (Figure R10). We believe that findings from this study will be of special interest to the readers of *Nature Communication*.

Furthermore, we would like to take this opportunity to further clarify our manuscript title “Lattice-Mismatch-Free Growth of Organic Heterostructure Nanowires: A General Large Scale Synthetic Strategy” for reviewer #2. Actually, we do not show in the title that a general strategy is applicable to all molecules, instead, the title of this work focuses on a general strategy for the large-scale synthesis of organic heterostructure nanowires, thus solving the main problem of lattice mismatch and phase separation during the co-assembly process of complex components. Undoubtedly, this strategy may not suit for all organic molecules due to the different organic molecules with unique carbon skeleton, but we can obtain the almost any heterostructure nanowires with distinct photoluminescence performances through choice suitable donors and acceptors via lattice-mismatch-free growth strategy, we think this methodology is general and effective.

Figure R6. (a) Scheme of the synthesis of the nine types of triblock nanowires composed of organic cocrystal and their alloys. (b-j) FM images of heterostructure nanowires under UV excitation. All scale bar is 10 μm .

Figure R7. (a-f) FM images of various AP cocrystal nanowires based on the Polycyclic Aromatic Hydrocarbons (PAH) and OFN molecules. (g-l) FM images of various CT cocrystal nanowires based on the PAH and TCPA molecules. All scale bar is 20 μm .

Figure R8. (a-f) Optical characterization of various heterostructures nanowires. (top image) FM image of individual triblock nanowire. The scale bar is 2 μm . (bottom image) PL spectra collected from different sections marked in (top image).

Figure R9. FM images of the large-scale synthesis of triblock nanowires

Figure R10. Molecular structures of organic heterostructure nanowires

2) very specific crystal shape which is the one of nanowires?

Response: We thank the reviewer's critical comments of our manuscript. Owing to the growth of nanowires along the one-dimensional direction, the heterostructure nanowires can be majorly divided into triblock, core/shell and branch types. Actually, we have realized a triblock and core/shell nanowire in manuscript (Figure 2 and Figure 4), which indicate that this strategy for heterostructure nanowires not only appears to one of nanowires. It is worth noting that this lattice-mismatch-free growth of organic heterostructure nanowires are implement on organic cocrystals and alloys with the same crystal structure and growth direction, thus forming the coaxial heterostructure nanowires such as triblock and core/shell. Therefore, we are convinced that this universal approach to construct triblock and core/shell heterostructures nanowires can also enlighten other organic heterostructure nanowires with any architectures composed

of organic cocrystal and alloys with different growth directions (vertical and lateral orientation) and lattice-matching facets.

3) The optical characterization and logic gate experiments have been already reported by the same authors and other in previous publications (for example ref. 43, 44).

Response: We greatly appreciate the reviewer's comments to help us strengthen our work. However, the reviewer thinks the work's optical characterization and logic gate mode have been already reported so it is not novel to perform the spatially resolved spectroscopy measurement and optical logic gates on heterostructure nanowires. That is unjust, for example (Refs 43, 44 *Nat. Commun.* **10**, 3839 (2019), <https://doi.org/10.1038/s41467-019-11731-7>; *CCS Chem.* **3**, 413 (2021), <https://doi.org/10.31635/ccschem.020.202000171>; *Nat. Commun.* **12**, 2252 (2021), <https://doi.org/10.1038/s41467-021-22513-5>), Zhuo et al. used optical characterization and optical logic gate to demonstrate optical property and application on nanowires but these experiments are also reported in previous article work (*J. Am. Chem. Soc.* **140**, 4269 (2018), <https://doi.org/10.1021/jacs.7b12519>; *Adv. Funct. Mater.* **28**, 1804915, (2018), <https://doi.org/10.1002/adfm.201804915>; *Adv. Funct. Mater.* **31**, 2105415, (2021), <https://doi.org/10.1002/adfm.202105415>; *Angew. Chem. Int. Ed.* **58**, 13890, (2019), <https://doi.org/10.1002/anie.201906278>; *Angew. Chem. Int. Ed.* **58**, 13803, (2019), <https://doi.org/10.1002/anie.201907433>.)! Conversely, we think that although optical applications (such as light harvesting, optical logic gate) has emerged as a hot topic in optoelectronics, the heterostructure materials is the significant foundation that can achieve this optical application. Therefore, the development of the synthesis of heterostructure nanowire materials should be emphasized. Notably, compared with the previous reports, several unique aspects of logic gate experiments on heterostructure nanowires of this work are worth highlighting up front:

- (i) the asymmetric optical signal output when excited at junction (Figure R11c, d), the one end emission blue color while another end emission olive color;
- (ii) the core/shell THSs have four coaxial output channels, which respectively located

on two tips and two junctions as shown in the Figure R11k, l;
(iii) the various heterostructure nanowires realize all-color-tunable photoluminescent performance, which can fulfill the different needs of optical signal processes.

The reviewer #2 also addressed several points in this revision:

1) Before resubmission to a more specialized journal, I urge the authors to revise the manuscript and try to have a more quantitative analysis of the optical data, for example by fitting the spectra with gaussian bands and report spectral shifts or spectral weights in a more quantitative and consistent way.

Response: We thank the reviewer's critical comments of our manuscript. We further use Matlab software to calculate the light intensity distribution of each pixel of the FM image for logic gates, and extract it to generate a three-dimensional image, where the z-axis value based on the light intensity (Figure R12). It can be seen that the z-axis values of input position keep the highest, while the z-axis value of the tips of nanowire is the middle due to the optical loss and self-absorption, which is coded as "signal 1", the low optical signal is defined "0". So that, we readily construct the optical logic gate with multiple input/output channels by Matlab calculation based on the light intensity of FM images of individual heterostructures nanowire under excitation at different position with a 375 nm laser. Take the absorption and PL spectrum of cocrystals, alloys and heterojunction as an example, we further perform a more quantitative analysis of the optical data, for example by fitting the spectra with gaussian bands and report spectral shifts or spectral weights in a more quantitative and consistent way (Figure R13 and R14). Actually, the multiple peaks of PL spectrum originate from the arene-perfluoroarene (AP) interaction cocrystals. (*Angew. Chem. Int. Ed.* **57**, 1928, (2018), <https://doi.org/10.1002/anie.201712104>; *Angew. Chem. Int. Ed.* **60**, 4575, (2021), <https://doi.org/10.1002/anie.202014891>; *Nat. Commun.* **10**, 169, (2019), <https://doi.org/10.1038/s41467-018-08092-y>). The single broad peak of PL spectrum originates from the charge-transfer (CT) interaction crystal (*Adv. Mater.*, **31**, 1902328 (2019), <https://doi.org/10.1002/adma.201902328>).

Figure R11. Photonic signal logic operation based on two types of triblock nanowire. (a, i) Schematic diagram of the triblock and core/shell triblock under a microarea excitation at two different positions with a 375 nm laser. (b-h) The corresponds to truth table and FM image, PL spectrum and CIE chromaticity diagram of individual blue-olive-blue heterostructure nanowire. (j-p) The corresponds to truth table and FM image, PL spectrum and CIE chromaticity diagram of individual orange-olive/orange-olive heterostructure nanowire.

Figure R12. (a) Schematic diagram of the individual triblock nanowire under excitation at two different positions with a 375 nm laser. (b) The logic table display intensity signal output's codes. (c, e) FM images at inputs I-1 and I-2. (d, f) The light intensity distribution of FM images obtained by Matlab simulation. (g) Schematic diagram of the individual core/shell triblock under excitation at two different positions with a 375 nm laser. (h) The logic table display intensity signal output's codes. (i, k) FM images at inputs I-1 and I-2 of core/shell nanowire. (j, l) The light intensity distribution of FM images (i, k) obtained by Matlab simulation.

Figure R13. The diffuse absorption and PL spectra of BGP-OFN, BGP-TFPA and BGP-OFN_(0.8)-TFPA_(0.2) alloy.

Figure R14. The PL spectra collected at the heterojunction of the blue-olive-blue nanowires by fitting the spectra with gaussian bands.

2) In the present version often the authors write about colors (blue, orange, olive sometimes called yellow). This is confusing and leaves space for interpretation.

Response: Thanks for the reviewer's careful examines and valuable comments. We have revised the description of colors in the manuscript.

Revision: On Page 9, line 12, some sentences have been revised in the manuscript. "The olive segment (3) shows a broad emission spectrum with single peak at 550 nm corresponding to the BGP-OFN_(0.8)-TFPA_(0.2) alloy emission." All the changes have been highlighted in the revised manuscript.

All the revisions and discussions can fully address the referees' comments and effectively enhance the quality of this work. We hope it can make sense and Reviewer #2 can reconsider our manuscript.

Thank you very much for your consideration.

REVIEWERS' COMMENTS

Reviewer #1 (Remarks to the Author):

Authors have addressed all the concerns I have raised and I am satisfied with their additional experiments and data analysis. I have no additional comments and suggestions. I am happy to accept the manuscript for publication.

Reviewer #2 (Remarks to the Author):

The authors have substantially improved the manuscript and now give details of an entire set of materials showing heterostructure growth. This is shown in Figure 6. I am positive about the publication of this work in Nature Communications. However I recommend the authors to specify in the title that they findings are valid for donor-acceptor or charge transfer co-crystals. I think this is very important to distinguish the materials of this study from the more general class of single component crystalline organic semiconductors.

RE: Revision for Manuscript ID NCOMMS-21-44837A-Z

Reviewer #1 (Remarks to the Author):

Authors have addressed all the concerns I have raised and I am satisfied with their additional experiments and data analysis. I have no additional comments and suggestions. I am happy to accept the manuscript for publication.

Response: We sincerely thank the reviewer for the kind evaluation on our work.

Reviewer #2 (Remarks to the Author):

The authors have substantially improved the manuscript and now give details of an entire set of materials showing heterostructure growth. This is shown in Figure 6.

I am positive about the publication of this work in Nature Communications. However I recommend the authors to specify in the title that they findings are valid for donor-acceptor or charge transfer co-crystals. I think this is very important to distinguish the materials of this study from the more general class of single component crystalline organic semiconductors.

Response: We sincerely thank the reviewer for the kind evaluation on our work. Thanks for the reviewer#2's suggestion. We have revised the title to make it more reasonable. We believe that the revised version is suitable for the publication in *Nature Communications*.

Revision: Title has been edited to "Lattice-mismatch-free growth of organic heterostructure nanowires from cocrystals to alloys".